# Hypothermia Inhibits Dexmedetomidine-Induced Contractions in Isolated Rat Aortae

**DOI:** 10.3390/ijms25053017

**Published:** 2024-03-05

**Authors:** Soohee Lee, Yeran Hwang, Kyeong-Eon Park, Sungil Bae, Seong-Ho Ok, Seung-Hyun Ahn, Gyujin Sim, Moonju Bae, Ju-Tae Sohn

**Affiliations:** 1Department of Anesthesiology and Pain Medicine, Gyeongsang National University Changwon Hospital, Changwon-si 51472, Gyeongsangnam-do, Republic of Koreamdoksh@naver.com (S.-H.O.); 2Department of Anesthesiology and Pain Medicine, Gyeongsang National University College of Medicine, Jinju-si 52727, Gyeongsangnam-do, Republic of Korea; 3Institute of Medical Science, Gyeongsang National University, Jinju-si 52727, Gyeongsangnam-do, Republic of Korea; 4Department of Anesthesiology and Pain Medicine, Gyeongsang National University College of Medicine, Gyeongsang National University Hospital, 15 Jinju-daero 816 Beon-gil, Jinju-si 52727, Gyeongsangnam-do, Republic of Korea; 5Department of Anesthesiology and Pain Medicine, Gyeongsang National University Hospital, 15 Jinju-daero 816 Beon-gil, Jinju-si 52727, Gyeongsangnam-do, Republic of Korea

**Keywords:** hypothermia, dexmedetomidine, contraction, calcium, myosin light chain

## Abstract

Dexmedetomidine is widely used to induce sedation in the perioperative period. This study examined the effect of hypothermia (33 and 25 °C) on dexmedetomidine-induced contraction in an endothelium-intact aorta with or without the nitric oxide synthase inhibitor N^W^-nitro-L-arginine methyl ester (L-NAME). In addition, the effect of hypothermia on the contraction induced by dexmedetomidine in an endothelium-denuded aorta with or without a calcium-free Krebs solution was examined. The effects of hypothermia on the protein kinase C (PKC), myosin light chain (MLC_20_) phosphorylation, and Rho-kinase membrane translocation induced by dexmedetomidine were examined. Hypothermia inhibited dexmedetomidine-induced contraction in the endothelium-intact aorta with L-NAME or endothelium-denuded aorta. Hypothermia had almost no effect on the dexmedetomidine-induced contraction in the endothelium-denuded aorta with the calcium-free Krebs solution; however, the subsequent contraction induced by the addition of calcium was inhibited by hypothermia. Conversely, the transition from profound hypothermia back to normothermia reversed the hypothermia-induced inhibition of subsequent calcium-induced contractions. Hypothermia inhibited any contraction induced by KCl, PDBu, and NaF, as well as PKC and MLC_20_ phosphorylation and Rho-kinase membrane translocation induced by dexmedetomidine. These results suggest that hypothermia inhibits dexmedetomidine-induced contraction, which is mediated mainly by the impediment of calcium influx and partially by the attenuation of pathways involving PKC and Rho-kinase activation.

## 1. Introduction

Highly selective alpha-2 adrenoceptor agonist dexmedetomidine (with a selective ratio of alpha-2 adrenoceptor/alpha-1 adrenoceptor in medetomidine: 1620) is widely used to induce sedation in the perioperative period [1,2]. Therapeutic hypothermia (such as mild and moderate hypothermia) can be applied as a treatment for traumatic brain injuries and following cardiac arrest [3,4]. Dexmedetomidine is used to sedate patients undergoing therapeutic hypothermia. However, hypothermia is associated with the following adverse effects: shivering, bradycardia, electrolyte imbalance, acute renal injury, coagulopathy, hypotension, and hypertension [5]. In addition, hypothermia causes endothelium-dependent vasodilation [6]. The intravenous administration of dexmedetomidine causes an initial transient increase in blood pressure, followed by a decrease [7]. In addition, a high-dose of dexmedetomidine leads to hypertension, which is possibly mediated by alpha-2 adrenoceptor activation in vascular smooth muscles [8,9,10,11]. Contractions in the vascular smooth muscle are mediated by the two following mechanisms: a calcium-dependent mechanism involving increased intracellular calcium levels and a calcium-sensitization mechanism associated with increased myofilament calcium sensitization [12]. This sensitization occurs through the inhibition of myosin light chain phosphatase induced by protein kinase C (PKC) and Rho-kinase [12]. Dexmedetomidine causes vasoconstriction, which is mediated by a calcium influx through voltage-operated calcium channels [13]. In addition, dexmedetomidine-induced contractions are mediated by calcium-sensitization through the pathways involving protein kinase C, Rho-kinase, and phosphoinositide 3-kinase [14]; however, this contraction is attenuated by endothelial nitric oxide [15]. Hypothermia induces nitric oxide-mediated endothelium-dependent vasodilation in isolated endothelium-intact arteries [6]. Furthermore, hypothermia attenuates vascular smooth muscle contraction induced by agonists, which is endothelium-independent and mediated by the inhibition of Rho-kinase and mitogen-activated protein kinase [16]. Therapeutic hypothermia, followed by rewarming, induces marked alterations in blood pressure [17]. However, although dexmedetomidine is used to sedate patients undergoing therapeutic hypothermia in intensive care units, the effects of hypothermia on dexmedetomidine-induced vascular responses remain unknown. Therefore, this study examined the effects and underlying mechanisms of hypothermia on the vascular response induced by dexmedetomidine in isolated rat aorta.

## 2. Results

Moderate hypothermia (33 °C) had no effect on dexmedetomidine (10^−6^ M)-induced contraction. However, it slightly inhibited the subsequent contraction induced by nitric oxide synthase inhibitor N^W^-nitro-L-arginine methyl ester (L-NAME, 10^−4^ M) in the endothelium-intact aorta treated with 10^−6^ M dexmedetomidine (Figure 1A; *p* < 0.001 vs. 37 °C) [5]. Profound hypothermia (25 °C) slightly inhibited dexmedetomidine (10^−6^ M)-induced maximal contraction (*p* < 0.001 vs. 37 °C); however, it markedly suppressed the subsequent contraction induced by L-NAME (10^−4^ M) in the endothelium-intact aorta treated with 10^−6^ M dexmedetomidine (Figure 1B: *p* < 0.001 vs. 37 °C) [5]. Moderate hypothermia inhibited dexmedetomidine-induced contractions in the endothelium-intact aorta pretreated with L-NAME (Figure 1C; *p* < 0.001 vs. 37 °C with L-NAME at 10^−7^ to 10^−6^ M). Profound hypothermia almost eliminated dexmedetomidine-induced contractions in the endothelium-intact aorta pretreated with L-NAME (Figure 1D: *p* < 0.001 vs. 37 °C with L-NAME at 10^−7^ to 10^−6^ M). Moderate hypothermia inhibited dexmedetomidine-induced contractions in the endothelium-denuded rat aorta (Figure 2A; *p* < 0.001 versus 37 °C at 3 × 10^−8^ to 10^−6^ M). Profound hypothermia almost eliminated dexmedetomidine-induced contractions in the endothelium-denuded rat aorta (Figure 2B; *p* < 0.001 vs. 37 °C at 3 × 10^−8^ to 10^−6^ M). However, the return to 37 °C from 25 °C eliminated the profound hypothermia-induced inhibition of contractions caused by dexmedetomidine (Figure 2B). Moderate and profound hypothermia had nearly no effect on dexmedetomidine (10^−9^ to 3 × 10^−7^ M)-induced contractions in the endothelium-denuded rat aorta with a calcium-free Krebs solution (Figure 3A,B). Moderate hypothermia slightly inhibited (Figure 3A; *p* < 0.001 vs. 37 °C) the subsequent contraction induced by calcium (1.25 and 2.5 mM). Moreover, profound hypothermia markedly inhibited the subsequent contraction induced by calcium (Figure 3B; *p* < 0.001 vs. 37 °C). However, the return to 37 °C from 25 °C eliminated the profound hypothermia-induced inhibition of contractions induced by the addition of calcium (1.25 and 2.5) in the calcium-free state (Figure 3B). Moderate hypothermia slightly inhibited 60 mM KCl-induced contractions in the endothelium-denuded rat aorta (Figure 4A; *p* < 0.001 vs. 37 °C at 10 to 80 min). Profound hypothermia greatly inhibited 60 mM KCl-induced contractions in the endothelium-denuded rat aorta (Figure 4B; *p* < 0.001 vs. 37 °C at 10 to 80 min). However, the return to 37 °C from 25 °C eliminated the profound hypothermia-induced inhibition of 60 mM KCl-induced contractions (Figure 4B at 40 min after 37 °C). Moderate hypothermia slightly inhibited Rho-kinase stimulant NaF (8 × 10^−3^ M)-induced contractions in the endothelium-denuded rat aorta (Figure 5A; *p* < 0.001 vs. 37 °C at 10 to 80 min). Profound hypothermia markedly inhibited NaF (8 × 10^−3^ M)-induced contractions in the endothelium-denuded rat aorta (Figure 5B; *p* < 0.001 vs. 37 °C at 10 to 80 min), but the return to 37 °C from 25 °C reversed the profound hypothermia-induced inhibition of contractions caused by NaF in the endothelium-denuded aorta (Figure 5B). Moderate hypothermia slightly inhibited PKC activator phorbol 12,13-dibutyrate (PDBu, 10^−6^ M)-induced contractions in the endothelium-denuded rat aorta (Figure 5C; *p* < 0.05 vs. 37 °C at 40 to 80 min). However, profound hypothermia markedly decreased PDBu-induced contractions (Figure 5D; *p* < 0.001 vs. 37 °C at 10 to 80 min). Nevertheless, the return to 37 °C from 25 °C eliminated the profound hypothermia-induced inhibition of contractions caused by PDBu (Figure 5D). The Rho-kinase inhibitor Y-27632 (3 × 10^−7^ to 3 × 10^−6^ M) and PKC inhibitor GF109203X (10^−6^ and 3 × 10^−6^ M) inhibited dexmedetomidine (10^−6^ M)-induced contractions (Figure 6A,B; *p* < 0.001 vs. time-matched control at 10^−6^ and 3 × 10^−6^ M).

Dexmedetomidine induced Rho-kinase membrane translocation and PKC phosphorylation (Figure 7; Rho-kinase membrane translocation: *p* < 0.01 versus control; PKC phosphorylation: *p* < 0.001 versus control). Profound hypothermia attenuated dexmedetomidine-induced Rho-kinase (ROCK-2) membrane translocation (Figure 7A; *p* < 0.01) and dexmedetomidine (10^−6^ M)-induced PKC phosphorylation (Figure 7B; *p* < 0.001). Moreover, dexmedetomidine (10^−6^ M) caused myosin light chain (MLC_20_) phosphorylation (Figure 7C; *p* < 0.001 vs. control); however, profound hypothermia inhibited dexmedetomidine-induced MLC_20_ phosphorylation (Figure 7C; *p* < 0.001).

## 3. Discussion

This study suggests that profound hypothermia inhibits the dexmedetomidine-induced contraction of vascular smooth muscle, which is mainly mediated by the inhibition of calcium influx from extracellular cellular space and partially by the inhibition of the calcium-sensitization pathway involving PKC and Rho-kinase. The major findings of this study are as follows: (1) Moderate and profound hypothermia inhibited dexmedetomidine-induced contractions in the endothelium-denuded rat aorta in a temperature-dependent manner. (2) Moderate and profound hypothermia had nearly no effect on dexmedetomidine-induced contractions in a calcium-free Krebs solution, but did inhibit subsequent calcium-induced contractions. However, raising the temperature to 37 °C from 25 °C eliminated the profound hypothermia-induced inhibition of calcium-induced contractions. (3) Hypothermia inhibited contractions induced by 60 mM KCl, NaF, and PDBu. (4) Profound hypothermia inhibited dexmedetomidine-induced PKC and MLC_20_ phosphorylation and Rho-kinase membrane translocation.

Calcium-dependent contractions of vascular smooth muscle are mediated by myosin light chain kinase activation and the subsequent phosphorylation of MLC_20_ through increased intracellular calcium levels [12]. An increase in the intracellular calcium level is mediated by calcium influx from extracellular space and calcium release from sarcoplasmic reticulum [12]. Dexmedetomidine-induced contractions are mediated by calcium influx through voltage-operated calcium channels [13]. In addition, dexmedetomidine-induced contractions are calcium-dependent, and hence, are completely inhibited in a calcium-free state [13]. In agreement with previous reports, dexmedetomidine-induced contractions were nearly eliminated in a calcium-free Krebs solution at 37 °C (Figure 3) [13]. As dexmedetomidine-induced contractions are inhibited by endothelial nitric oxide, we examined the effect of endothelial nitric oxide synthase inhibitor L-NAME on the hypothermia-mediated inhibition of dexmedetomidine-induced contractions in the endothelium-intact aorta [15]. In a previous study, hypothermia (28 °C) inhibited agonist-induced contractions in vascular smooth muscle, which is endothelium-independent [16]. Similarly, in our study, moderate and profound hypothermia inhibited dexmedetomidine-induced contractions in the endothelium-intact aorta which had been pretreated with L-NAME (Figure 1C,D), and profound hypothermia severely inhibited subsequent contractions induced by the addition of L-NAME into the endothelium-intact aorta demonstrating dexmedetomidine-induced contractions (Figure 1B) [16]. This suggests that the hypothermia-mediated inhibition of dexmedetomidine-induced contractions is independent of the endothelial nitric oxide. Furthermore, moderate and profound hypothermia inhibited dexmedetomidine-induced contractions in the endothelium-denuded rat aorta in a temperature-dependent manner (Figure 2A,B). This indicates that the hypothermia-mediated inhibition of dexmedetomidine-induced contractions is independent of the endothelium. In addition, moderate and profound hypothermia inhibited contractions induced by the addition of calcium into the endothelium-denuded rat aorta showing dexmedetomidine (10^−6^ M)-induced contractions in a calcium-free state (Figure 3A,B), which seem to be temperature-dependent. However, the raising of the temperature to 37 °C almost reversed the hypothermia-induced inhibition of contractions caused by dexmedetomidine (Figure 3A,B). Taken together, these results suggest that the hypothermia-mediated inhibition of dexmedetomidine-induced contractions occurs through the inhibition of calcium influx. As dexmedetomidine-induced contractions are dependent on the concentration of calcium entering through voltage-operated calcium channels, we examined the effect of hypothermia on contractions induced by 60 mM KCl, which are mediated by calcium influx through voltage-operated calcium channels [12,13]. Consistent with the aforementioned observations, both moderate and profound hypothermia inhibited 60 mM KCl-induced contractions in a temperature-dependent manner (Figure 4). However, the transition from 25 °C to 37 °C reversed the profound hypothermia-induced inhibition of 60 mM KCl-induced contractions. This suggested that hypothermia inhibited the contraction via the impeding of calcium influx through voltage-operated calcium channels.

The calcium-sensitization-mediated contraction of vascular smooth muscle is induced by the inhibition of myosin light chain phosphatase and the subsequently increased phosphorylation of MLC_20_ [12]. The inhibition of myosin light chain phosphatase is mediated by the phosphorylation of the myosin phosphatase target subunit or the phosphorylation-dependent inhibitory protein of myosin phosphatase, which is induced by PKC and Rho-kinase [12]. Calcium sensitization associated with dexmedetomidine-induced contractions is mediated by the phosphorylation-dependent inhibitory protein of myosin phosphatase, which is activated by PKC and Rho-kinase [18]. Similarly, Rho-kinase inhibitor Y-27632 and PKC inhibitor GF1092203X inhibited dexmedetomidine (10^−6^ M)-induced contractions in endothelium-denuded rat aorta, suggesting that dexmedetomidine-induced contractions are mediated by PKC and Rho-kinase [18]. In agreement with the results of a current tension study, dexmedetomidine in vascular smooth muscle cells induced PKC phosphorylation and Rho-kinase membrane translocation. However, profound hypothermia (25 °C) inhibited PKC phosphorylation and Rho-kinase membrane translocation induced by dexmedetomidine, suggesting that hypothermia attenuated the pathways associated with the dexmedetomidine-induced activation of PKC and Rho-kinase. Therefore, given that NaF and PDBu activates Rho-kinase and PKC, respectively, we assessed the effect of hypothermia on contractions induced by NaF and PDBu to confirm if hypothermia attenuates pathways associated with the roles of PKC and Rho-kinase in vascular smooth muscle contractions [19,20]. In agreement with the results of Western blot testing and a previous report, hypothermia attenuated contractions induced by NaF and PDBu, supporting the idea that hypothermia inhibited the activation of pathways involving PKC and Rho-kinase in contractions [16]. As dexmedetomidine-induced contractions are calcium-dependent, dexmedetomidine-induced MLC_20_ phosphorylation observed in the current study is seemingly attributable to the increased calcium-induced activation of myosin light chain kinase [12,13]. Moreover, profound hypothermia inhibited the contraction induced by the addition of calcium in the calcium-free state with 10^−6^ M dexmedetomidine-induced contractions (Figure 3B), suggesting that hypothermia inhibits calcium influx from the extracellular space. Taken together, the results suggest that the profound hypothermia-induced inhibition of MLC_20_ phosphorylation which is triggered by dexmedetomidine may be attributed to the hypothermia-induced inhibition of calcium influx. Further studies should explore the detailed upstream cellular signal pathway associated with the inhibition of MLC_20_ phosphorylation by dexmedetomidine under hypothermic conditions.

The limitations of this study are as follows: First, blood pressure is determined by cardiac output and peripheral vascular resistance. Therefore, peripheral vascular resistance is mainly determined by small resistance arterioles [21]. However, this study used the aorta of a rat, which is regarded as a conduit vessel. Second, hypothermia (core temperature: 34 °C) produces decreased cardiac output induced by bradycardia and reflex vasoconstriction in skin in vivo [22]. However, this study is an in vitro study using isolated rat aortae. Even with these limitations, hypothermia, which is employed for neuroprotection or post-cardiac arrest resuscitation in intensive care units, may attenuate hypertension induced by an inadvertent high dose of dexmedetomidine used for sedation [8,9,10]. In addition, as dexmedetomidine-induced contractions are attenuated by endothelial nitric oxide release and increased by nitric oxide synthase inhibitor L-NAME, dexmedetomidine-induced contractions may be increased in patients with compromised endothelium, such as those with hypertension, diabetes, or atherosclerosis [15]. Therefore, hypothermia may attenuate hypertension caused by dexmedetomidine-induced exaggerated vasoconstriction in patients with compromised endothelia.

## 4. Materials and Methods

The research protocol (GNU-231030-R0199) received approval from the Institutional Animal Care and Use Committee at the Gyeongsang National University. Our study was conducted in accordance with the established Animal Care and Use guidelines. A male Sprague–Dawley rat weighing 250–280 g, obtained from Koatech (Pyeongtaek, Republic of Korea), was rendered unconscious using 100% CO_2_ as the anesthesia. To evaluate isometric tension, the isolated rat aorta was prepared using the method described in a prior study [23]. The rat’s chest was opened, and the descending thoracic aorta was removed from the thoracic cage. The removed thoracic aorta was placed in a Krebs solution that comprised sodium chloride (118 mM), sodium bicarbonate (25 mM), glucose (11 mM), potassium chloride (4.7 mM), calcium chloride (2.4 mM), magnesium sulfate (1.2 mM), and monopotassium phosphate (1.2 mM). The connective tissue and adipose (fat) surrounding the separated rat aorta within the Krebs solution were meticulously removed under a microscope. Subsequently, the isolated descending thoracic aorta was divided into segments measuring 2.5 mm in length. In some aortae, the inner endothelial layer was removed by gently rolling the aorta back and forth using two 25-gauge needles inserted into the aortic lumen. Next, the isolated descending thoracic rat aorta was suspended within an organ bath equipped with a Grass isometric transducer (FT-03, Grass Instrument, Quincy, MA, USA), and the bath was maintained at a constant temperature of 37 °C. As outlined in an earlier study, a starting resting tension of 24.5 mN was sustained for a duration of 1.5 h in order to reach a stable plateau [24]. Throughout this period, the original Krebs solution was replaced with fresh Krebs solution every 30 min. The pH of the Krebs solution was kept at 7.4 by continually supplying it with a mixture of 95% O_2_ and 5% CO_2_. To assess the integrity of the endothelium in the rat aorta that still had its inner lining intact, the following procedure was carried out [23]: Following the induction of a sustained and consistent contraction with phenylephrine (10^−7^ M), acetylcholine (10^−5^ M) was introduced into the organ bath. If the aorta displayed over 85% relaxation in response to acetylcholine from the previous phenylephrine-induced contraction, then it was categorized as an endothelium-intact aorta. Endothelial denudation was confirmed following a previously reported procedure [23]. After phenylephrine (10^−8^ M) had produced a persistent and stable aortic contraction, acetylcholine (10^−5^ M) was introduced into the organ bath. An aorta that exhibited <15% relaxation in response to acetylcholine was classified as an endothelium-denuded aorta. Rat aortae with their endothelial lining preserved and those without endothelia were rinsed multiple times to return them to their initial resting tension levels. Following this, the subsequent experimental procedures were conducted: To quantify the relative magnitude of dexmedetomidine (10^−9^ to 10^−6^ M)-induced contractions, isotonic 60 mM KCl-induced contractions were measured in some aortae with or without endothelia. After inducing contractions with isometric 60 mM KCl, the system was thoroughly washed with a fresh Krebs solution to return to the baseline resting tension. Subsequent experimental procedures were then carried out according to the specified protocols.

### 4.1. Experiment Protocols

First, we investigated the effect of hypothermia on the contraction induced by dexmedetomidine (10^−9^ to 10^−6^ M) and any subsequent contractions triggered by nitric oxide synthase inhibitor L-NAME (10^−4^ M) in endothelium-intact aorta without prior treatment with 10^−4^ M L-NAME. Some endothelium-intact rat aortae were pretreated with 10^−4^ M L-NAME for 20 min. Subsequently, endothelium-intact rat aortae, with or without pretreatment with 10^−4^ M L-NAME, were subjected to hypothermia (33 °C [moderate] or 25 °C [profound]) for 40 min. Dexmedetomidine (10^−9^ to 10^−6^ M) was then cumulatively added into the organ bath to induce contractions at 25, 33, and 37 °C. Additionally, following the induction of maximal contractions using dexmedetomidine (10^−6^ M) in endothelium-intact aorta without pretreatment with L-NAME at 25, 33, and 37 °C, L-NAME (10^−4^ M) was added to the organ bath to trigger L-NAME-induced contractions in the endothelium-intact aorta already experiencing 10^−6^ M dexmedetomidine-induced contractions.

Second, the effects of hypothermia on the dexmedetomidine-induced contractions in the endothelium-denuded aorta were examined under two conditions: with or without pretreatment with a calcium-free Krebs solution. After some endothelium-denuded rat aortae without or with a calcium-free Krebs solution were exposed to hypothermic temperatures (33 or 25 °C) for 40 min, dexmedetomidine (10^−9^ to 10^−6^ M) was cumulatively added to the organ bath in order to generate dexmedetomidine concentration–response curves in the presence or absence of hypothermia. After dexmedetomidine (10^−6^ M) induced maximal contractions in the endothelium-denuded rat aorta pretreated with a calcium-free Krebs solution, calcium (1.25 and 2.5 mM) was cumulatively added to the calcium-free Krebs solution to achieve calcium-induced maximal contractions in the presence or absence of hypothermia (33 or 25 °C). Subsequently, the temperature was raised from hypothermia (33 or 25 °C) to 37 °C to assess the effect of normal temperatures on the dexmedetomidine (10^−6^ M)-induced contractions in the Krebs solution or the calcium (2.5 mM)-induced contractions in the endothelium-denuded rat aorta which were exposed to dexmedetomidine (10^−6^ M) in the calcium-free Krebs solution.

Third, the effects of hypothermia on the contraction induced by voltage-operated calcium channel agonist KCl, Rho-kinase activator NaF, and PKC activator PDBu in endothelium-denuded rat aortae were examined [19,20]. The 60 mM KCl-induced contraction in the endothelium-denuded rat aorta was sustained at 37 °C, and then monitored for 80 min in the presence or absence of hypothermia (33 or 25 °C). Next, the temperature was raised to 37 °C from hypothermia (33 and 25 °C) to examine the effect of normal temperatures (37 °C) on the contraction induced by 60 mM KCl in the hypothermic state (33 or 25 °C). In addition, after the contraction induced by NaF (8 × 10^−3^ M) and PDBu (10^−6^ M) in the endothelium-denuded rat aorta pretreated with L-NAME (10^−4^ M) was sustained at 37 °C, it was monitored for 80 min in the presence or absence of hypothermia (33 or 25 °C). Next, the temperature was raised to 37 °C from 25 °C to examine the effect of normal temperatures on the contraction induced by NaF or PDBu in the hypothermic state.

Fourth, the effects of Rho-kinase inhibitor Y-27632 and PKC inhibitor GF109203X on the dexmedetomidine-induced contractions in the endothelium-denuded rat aorta were examined. After dexmedetomidine (10^−6^ M) produced sustained and stable contractions in the endothelium-denuded rat aorta, Y-27632 (3 × 10^−7^ to 3 × 10^−6^ M) or GF109203X (3 × 10^−7^ to 3 × 10^−6^ M) was cumulatively added to the organ bath in order to examine the effect of Rho-kinase or the PKC inhibitor on the dexmedetomidine-induced contraction.

### 4.2. Cell Culture

Vascular smooth muscle cells were isolated from the descending thoracic aorta and cultured in Dulbecco’s modified Eagle’s medium (Gibco, Life Technologies, Grand Island, NY, USA). This medium was enriched with 100 μg/mL of streptomycin, 100 U/mL of penicillin, and 10% fetal bovine serum, as outlined in a prior study [25]. Cells from passages 3 to 5 were employed for this study, and they were maintained in a controlled environment at 37 °C with 5% carbon dioxide in humid air. Vascular smooth muscle cells were subjected to a serum-free medium for 15 h prior to the administration of the drug.

### 4.3. Western Blot

Western blot analysis was conducted following the procedure of Lee et al. [25]. Cells cultivated in 100 mm or 150 mm dishes were exposed to dexmedetomidine. Cells were treated with dexmedetomidine (10^−6^ M) alone at 37 °C for 30 min, hypothermia (25 °C) for 30 min followed by dexmedetomidine (10^−6^ M) for 30 min, or hypothermia (25 °C) alone for 60 min to assess ROCK-2 membrane translocation. To determine the expression of PKC phosphorylation, cells were treated with dexmedetomidine (10^−6^ M) alone at 37 °C for 10 min, hypothermia (25 °C) for 30 min followed by dexmedetomidine (10^−6^ M) for 10 min, or hypothermia (25 °C) alone for 40 min. In addition, cells were treated with dexmedetomidine (10^−6^ M) alone at 37 °C for 20 min, hypothermia (25 °C) for 30 min followed by dexmedetomidine (10^−6^ M) for 20 min, or hypothermia (25 °C) alone for 50 min to determine the expression of MLC_20_ phosphorylation. Subsequently, they were rinsed twice with a phosphate-buffered saline. In preparation for the immunoblot analysis, complete cell lysis was achieved by using radio-immunoprecipitation assay buffers (Cell Signaling Technology in Beverly, MA, USA). This process was employed to acquire total cell lysates. The cell lysates were subjected to centrifugation at 20,000× *g* for 15 min at 4 °C. Afterward, the resulting supernatants, containing 30 µg of protein, were boiled for 10 min. Next, the proteins were separated using 8–15% sodium dodecyl sulfate-polyacrylamide gel electrophoresis. The separated proteins were moved onto polyvinylidene difluoride membranes. After blocking with a 5% solution of bovine serum albumin or 5% skim milk in Tris-buffered saline with 0.5% Tween-20 (TBST) at room temperature (23 to 27 °C) for 1 h, the membranes were subjected to overnight incubation at 4 °C with primary antibodies, specifically anti-ROCK-2 (diluted 1:500), anti-PKC (diluted 1:200), anti-phospho-PKC (diluted 1:2000), anti-MLC_20_ (diluted 1:1000), and anti-phospho-MLC_20_ (diluted 1:500). Following a wash with TBST, the membranes were exposed to horseradish peroxidase-conjugated anti-rabbit or anti-mouse IgG antibodies, which were diluted at a ratio of 1:5000 in TBST containing 5% skim milk, for 1 h at room temperature. The immune complexes were identified using the Westernbright^TM^ ECL Western blotting kit (Advansta, San Jose, CA, USA). The density values of the bands were determined using ImageJ software (version 1.45s; National Institutes of Health, Bethesda, MD, USA).

### 4.4. Materials

We used pharmaceuticals and chemicals that met the highest purity standards and were readily available in the market. Dexmedetomidine, NaF, GF109203X, L-NAME, and PDBu were purchased from Sigma-Aldrich (St. Louis, MO, USA). Anti-PKC, anti-phospho-PKC, anti-MLC_20_, and anti-phopho-MLC_20_ antibodies were purchased from Cell Signaling Technology (Beverly, MA, USA). Y-27632 was provided by Calbiochem (La Jolla, CA, USA). Anti-ROCK-2 antibody was purchased from Santa Cruz (CA, USA). PDBu and GF109203X were dissolved in dimethyl sulfoxide, with the final concentration of dimethyl sulfoxide being <0.1%. Unless specified otherwise, all medications were dissolved and thinned with distilled water.

### 4.5. Statistical Analysis

The effects of hypothermia and inhibitors on the dexmedetomidine-induced contractions were analyzed using a linear mixed effect model (Stata version 14.2, StataCorp LLC, Lakeway Drive, College Station, TX, USA) [26]. In addition, the effects of hypothermia on the contraction induced by KCl, NaF, and PDBu were analyzed using a linear mixed effect model. The normality test was performed using the Kolmogorov–Smirnov test. The effect of hypothermia on PKC and MLC_20_ phosphorylation and Rho-kinase membrane translocation induced by dexmedetomidine was analyzed using a one-way analysis of variance, followed by Bonferroni’s test with Prism 5.0 (GraphPad Inc., San Diego, CA, USA). A *p* value < 0.05 indicated statistical significance.

## 5. Conclusions

The results of this study suggest that hypothermia attenuates the dexmedetomidine-induced contractions of vascular smooth muscle, which is mediated mainly by the inhibition of calcium influx and partially by the suppression of pathways involving PKC and Rho-kinase activation.

## Figures and Tables

**Figure 1 ijms-25-03017-f001:**
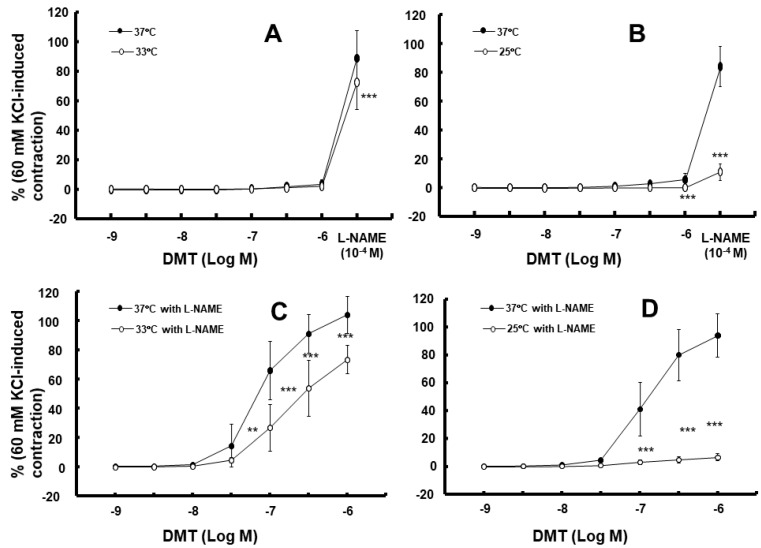
Effect of hypothermia (33 and 25 °C) on the dexmedetomidine (DMT)-induced contraction in the endothelium-intact rat aorta without (**A**,**B**) or with (**C**,**D**) N^W^-nitro-L-arginine methyl ester (L-NAME, 10^−4^ M). L-NAME (10^−4^ M) was added to the organ bath after the 10^−6^ M DMT-induced maximal contraction of the endothelium-intact aorta without pretreatment with L-NAME. Data (N = 5) are shown as mean ± standard deviation and expressed as the percentage of contractions induced by 60 mM KCl. N indicates the number of rats from which thoracic aortae were obtained. ** *p* < 0.01 and *** *p* < 0.001 vs. 37 °C or 37 °C with L-NAME.

**Figure 2 ijms-25-03017-f002:**
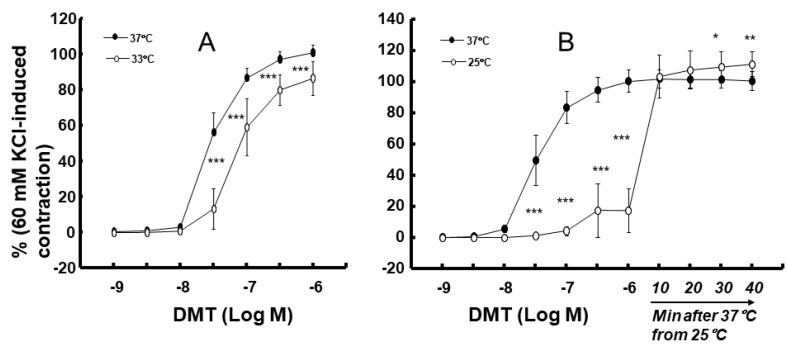
Effect of hypothermia (33 (**A**) and 25 (**B**) °C) on the dexmedetomidine (DMT)-induced contraction in endothelium-denuded rat aorta. The temperature in the 25 °C hypothermia group was raised to 37 °C after the 10^−6^ M DMT-induced maximal contraction. Data (33 °C: N = 5, and 25 °C: N = 6) are shown as mean ± standard deviation and expressed as the percentage of contractions induced by 60 mM KCl. N indicates the number of rats from which thoracic aortae were obtained. * *p* < 0.05, ** *p* < 0.01, and *** *p* < 0.001 vs. 37 °C.

**Figure 3 ijms-25-03017-f003:**
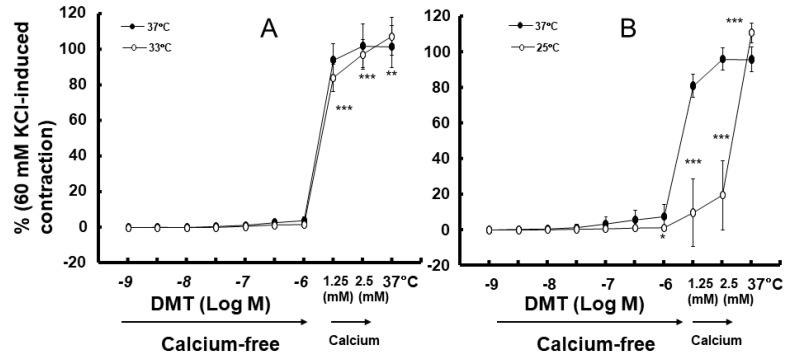
Effect of hypothermia (33 (**A**) and 25 (**B**) °C) on the dexmedetomidine (DMT)-induced contraction in the calcium-free Krebs solution and the subsequent calcium-induced contraction. After the calcium (2.5 mM)-induced contraction in the hypothermia group (33 or 25 °C), the temperature increased to 37 °C. Data (N = 5) are shown as mean ± standard deviation and expressed as the percentage of contractions induced by 60 mM KCl. N indicates the number of rats from which thoracic aortae were obtained. * *p* < 0.05, ** *p* < 0.01, and *** *p* < 0.001 vs. 37 °C.

**Figure 4 ijms-25-03017-f004:**
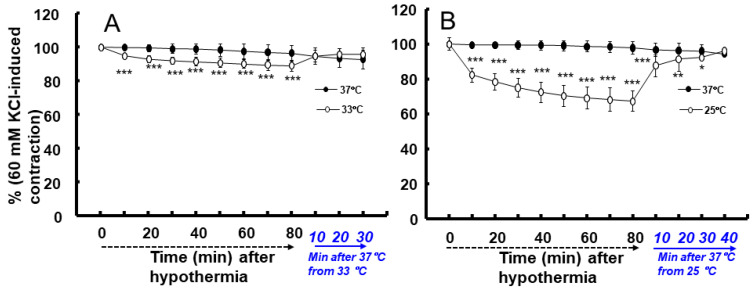
Effect of hypothermia (33 (**A**) and 25 (**B**) °C) on the contraction induced by 60 mM KCl in endothelium-denuded rat aorta. The temperature in the hypothermia group was raised to 37 °C, following the monitoring of the 60 mM KCl-induced contraction for 80 min following hypothermia. Data (N = 6) are shown as mean ± standard deviation and expressed as the percentage of contractions induced by 60 mM KCl. N indicates the number of rat from which thoracic aortae were obtained. * *p* < 0.05, ** *p* < 0.01, and *** *p* < 0.001 vs. 37 °C.

**Figure 5 ijms-25-03017-f005:**
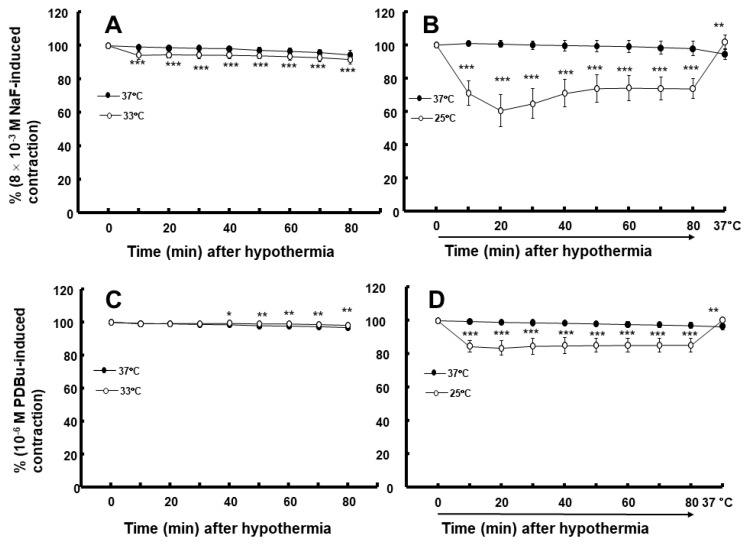
Effect of hypothermia (33 and 25 °C) on the contraction induced by NaF (8 × 10^−3^ M: (**A**) [N = 5] and (**B**) [N = 7]) and phorbol 12,13-dibutyrate (PDBu, 10^−6^ M: (**C**) [N = 5] and (**D**) [N = 5]) in endothelium-denuded rat aortae. The temperature in the 25 °C hypothermia group was raised to 37 °C after monitoring the contraction induced by NaF or PDBu for 80 min, following profound hypothermia (25 °C). Data are shown as mean ± standard deviation and expressed as the percentage of contractions induced by PDBu or NaF. N indicates the number of rats from which thoracic aortae were obtained. * *p* < 0.05, ** *p* < 0.01, and *** *p* < 0.001 vs. 37 °C.

**Figure 6 ijms-25-03017-f006:**
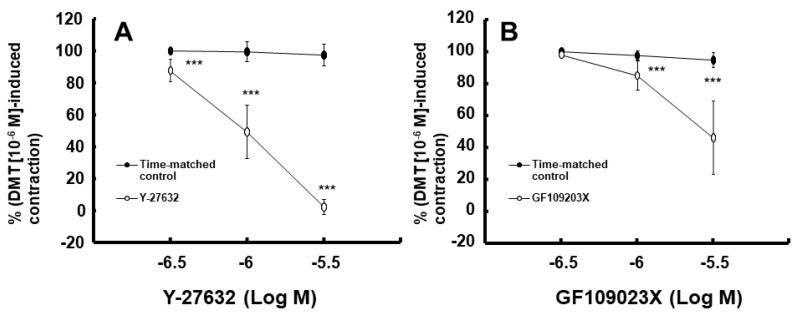
Effect of Y-27632 (**A**) and GF109203X (**B**) on the contraction induced by dexmedetomidine (DMT, 10^−6^ M) in endothelium-denuded rat aorta. Data (N = 5) are shown as mean ± standard deviation and expressed as the percentage of contractions induced by DMT (10^−6^ M). N indicates the number of rats from which thoracic aortae were obtained. *** *p* < 0.001 vs. time-matched control.

**Figure 7 ijms-25-03017-f007:**
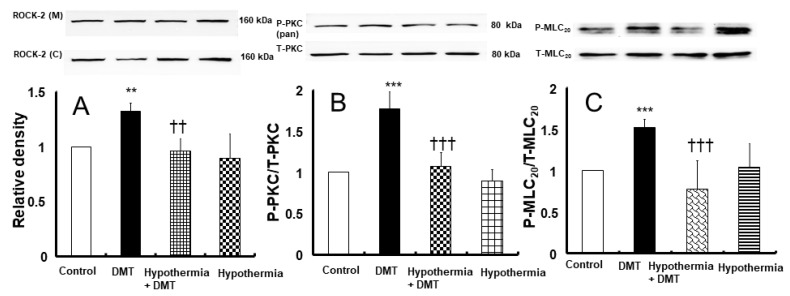
Effect of hypothermia (25 °C) on Rho-kinase (ROCK-2) membrane translocation ((**A**); N = 4), protein kinase C (PKC, (**B**); N = 5), and myosin light chain (MLC_20_, (**C**); N = 5) phosphorylation induced by dexmedetomidine (DMT, 10^−6^ M) in rat aortic vascular smooth muscle cells. Data are shown as mean ± standard deviation. N indicates the number of independent experiments. ** *p* < 0.01 and *** *p* < 0.001 vs. control. †† *p* < 0.01 and ††† *p* < 0.001 vs. DMT alone. ROCK-2 (M): ROCK-2 (membrane), ROCK-2 (C): ROCK-2 (cytosol), T-PKC: total PKC, P-PKC (pan): phosphorylated PKC (pan), T-MLC_20_: total MLC_20_, P-MLC_20_: phosphorylated MLC_20_.

## Data Availability

The data presented in this study are available upon reasonable request.

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
