# Peer review of "Hypothermia Inhibits Dexmedetomidine-Induced Contractions in Isolated Rat Aortae"

_ijms, 2024, doi:10.3390/ijms25053017_

Round 1

Reviewer 1 Report

Comments and Suggestions for Authors

The authors studied the effect of hypothermia on dexmedetomidine-induced contraction in endothelium-intact aortas with or without nitric oxide synthase inhibitor NW-nitro-L-arginine methyl ester and in endothelium-depleted aortas without or with Krebs solution without calcium. The results showed that hypothermia inhibits dexmedetomidine-induced contraction in the aorta with intact endothelium with L-NAME or in the aorta without endothelium; that hypothermia had almost no effect on dexmedetomidine-induced contraction in endothelium-free aorta with calcium-free Krebs solution, and that subsequent contraction induced by calcium addition was inhibited by hypothermia. These results suggested that hypothermia inhibited dexmedetomidine-induced contraction, which was mediated primarily by impeding calcium influx and partially by attenuating pathways involving PKC and Rho-kinase activation.

The study is well conducted, and the results are the beginning of new information.

The graphs are clear and the bibliography correct.

I suggest reviewing a few English errors probably due to typos.

Comments on the Quality of English Language

I suggest reviewing a few English errors probably due to typos.

Author Response

Response to Reviewer #1’s comments

Thank you very much for your valuable comments.

The authors studied the effect of hypothermia on dexmedetomidine-induced contraction in endothelium-intact aortas with or without nitric oxide synthase inhibitor NW-nitro-L-arginine methyl ester and in endothelium-depleted aortas without or with Krebs solution without calcium. The results showed that hypothermia inhibits dexmedetomidine-induced contraction in the aorta with intact endothelium with L-NAME or in the aorta without endothelium; that hypothermia had almost no effect on dexmedetomidine-induced contraction in endothelium-free aorta with calcium-free Krebs solution, and that subsequent contraction induced by calcium addition was inhibited by hypothermia. These results suggested that hypothermia inhibited dexmedetomidine-induced contraction, which was mediated primarily by impeding calcium influx and partially by attenuating pathways involving PKC and Rho-kinase activation.

The study is well conducted, and the results are the beginning of new information.

The graphs are clear and the bibliography correct.

I suggest reviewing a few English errors probably due to typos.

Response:

The English of this revised manuscript was reviewed by a native English speaker from Editage.

Reviewer 2 Report

Comments and Suggestions for Authors

The author effectively elucidated how hypothermia hinders the contraction induced by dexmedetomidine in the isolated rat aorta. I have full confidence in the study.

I observed a similarity in another published article titled "Dexmedetomidine-induced contraction of isolated rat aorta is dependent on extracellular calcium concentration" in the Korean Journal of Anesthesiology (2012 Sep; 63(3): 253–259). Could you clarify the differences between these two papers?

Author Response

Response to Reviewer #2’s comments

Thank you very much for your valuable comments.

The author effectively elucidated how hypothermia hinders the contraction induced by dexmedetomidine in the isolated rat aorta. I have full confidence in the study.

I observed a similarity in another published article titled "Dexmedetomidine-induced contraction of isolated rat aorta is dependent on extracellular calcium concentration" in the Korean Journal of Anesthesiology (2012 Sep; 63(3): 253–259). Could you clarify the differences between these two papers?

Response:

Thank you very much.

The previous report entitled “Dexmedetomidine-induced contraction of isolated rat aorta is dependent on extracellular calcium concentration” examined the underlying mechanisms associated with dexmedetomidine-induced contraction in isolated endothelium-denuded rat aortae using only isometric tension measurement. However, our current study examined the effect of hypothermia on the contraction induced by dexmedetomidine in isolated endothelium-intact and -denuded rat aortae using isometric tension measurement and western blotting and the underlying mechanism. In addition, the effect of hypothermia on the contraction induced by voltage-operated calcium channel agonist KCl, Rho-kinase activator NaF, and PKC stimulant PDBu was examined to elucidate the underlying mechanism associated with hypothermia-mediated inhibition of dexmedetomidine-induced contraction. Furthermore, the effect of hypothermia on the myosin light chain (MLC20) and PKC phosphorylation, Rho-kinase membrane translocation induced by dexmedetomidine was examined to elucidate the involvement of the calcium-dependent and -sensitization mechanism associated with dexmedetomidine-induced contraction. Thus, this study offers a comprehensive examination of the effect of hypothermia on contraction induced by alpha-2 adrenoceptor agonist dexmedetomidine and the underlying mechanism.  
